# Epidemiological Characteristics and Survival in Patients with De Novo Metastatic Prostate Cancer

**DOI:** 10.3390/cancers12102855

**Published:** 2020-10-03

**Authors:** Carlo Cattrini, Davide Soldato, Alessandra Rubagotti, Linda Zinoli, Elisa Zanardi, Paola Barboro, Carlo Messina, Elena Castro, David Olmos, Francesco Boccardo

**Affiliations:** 1Department of Internal Medicine and Medical Specialties (DIMI), School of Medicine, University of Genoa, 16132 Genoa, Italy; davide.soldato@gmail.com (D.S.); elisa.zanardi@unige.it (E.Z.); fboccardo@unige.it (F.B.); 2Prostate Cancer Clinical Research Unit, Spanish National Cancer Research Centre (CNIO), 28029 Madrid, Spain; dolmos@cnio.es; 3Academic Unit of Medical Oncology, IRCCS Ospedale Policlinico San Martino, 16132 Genoa, Italy; alessandra.rubagotti@unige.it (A.R.); datamanager.omb@unige.it (L.Z.); paola.barboro@hsanmartino.it (P.B.); 4Department of Health Sciences (DISSAL), School of Medicine, University of Genoa, 16132 Genoa, Italy; 5Department of Medical Oncology, Santa Chiara Hospital, 38122 Trento, Italy; carlo.messina@apss.tn.it; 6CNIO-IBIMA Genitourinary Cancer Unit, Hospitales Universitarios Virgen de la Victoria y Regional de Málaga, Instituto de Investigación Biomédica de Málaga, 29010 Malaga, Spain; ecastro@ext.cnio.es

**Keywords:** prostatic neoplasms/mortality, prostatic neoplasms/epidemiology, SEER Program

## Abstract

**Simple Summary:**

In randomized trials, both chemotherapy and androgen-receptor signaling inhibitors provided significant survival benefits in patients with metastatic prostate cancer (mPCa). However, it is largely unknown to what extent these therapeutic advances have impacted the general, real-world survival of patients with de novo mPCa. Here, we analyzed more than 26,000 patients included in the U.S. Surveillance, Epidemiology, and End Results (SEER) database to describe potential recent improvements in overall and cancer-specific survival. We found that patients diagnosed in the latest years showed a modest reduction in the risk of death and cancer-specific death, compared with those diagnosed in 2000–2003 and 2004–2010. Although our analysis was not adjusted for many confounders, the overall population of patients diagnosed in 2011–2014 only showed a survival gain of 4 months. Patients’ ineligibility or refusal of anticancer treatments, insurance issues, intrinsic disease aggressiveness, or prior unavailability of drugs in a hormone-sensitive setting might contribute to these disappointing results.

**Abstract:**

The real-world outcomes of patients with metastatic prostate cancer (mPCa) are largely unexplored. We investigated the trends in overall survival (OS) and cancer-specific survival (CSS) in patients with de novo mPCa according to distinct time periods. The U.S. Surveillance, Epidemiology, and End Results (SEER) Research Data (2000–2017) were analyzed using the SEER*Stat software. The Kaplan–Meier method and Cox regression were used. Patients with de novo mPCa were allocated to three cohorts based on the year of diagnosis: A (2000–2003), B (2004–2010), and C (2011–2014). The maximum follow-up was fixed to 5 years. Overall, 26,434 patients were included. Age, race, and metastatic stage (M1) significantly affected OS and CSS. After adjustment for age and race, patients in Cohort C showed a 9% reduced risk of death (hazard ratio (HR): 0.91 (95% confidence interval [CI] 0.87–0.95), *p* < 0.001) and an 8% reduced risk of cancer-specific death (HR: 0.92 (95% CI 0.88–0.96), *p* < 0.001) compared with those in Cohort A. After adjustment for age, race, and metastatic stage, patients in Cohort C showed an improvement in OS and CSS compared with Cohort B (HR: 0.94 (95% CI 0.91–0.97), *p* = 0.001; HR: 0.89 (95% CI 0.85–0.92), *p* < 0.001). Patients with M1c disease had a more pronounced improvement in OS and CSS compared with the other stages. No differences were found between Cohorts B and C. In conclusion, the real-world survival of de novo mPCa remains poor, with a median OS and CSS improvement of only 4 months in the latest years.

## 1. Introduction

The treatment landscape of metastatic prostate cancer (mPCa) has completely changed over the last decades. In 2004, docetaxel was the first drug to demonstrate an overall survival (OS) benefit of 2.4 months in mPCa, compared with mitoxantrone, and was approved for the treatment of men with metastatic castration-resistant prostate cancer (mCRPC) [1]. Cabazitaxel showed a similar OS increase compared with mitoxantrone and became a second-line treatment option for mCRPC in 2010 [2]. Subsequently, abiraterone acetate and enzalutamide were approved in both post-docetaxel [3,4] (2011–2012) and pre-docetaxel mCRPC [5,6] (2013–2014), reporting OS advantages between 4.0 and 4.8 months compared with placebo (Figure 1). Docetaxel was also introduced for the hormone-sensitive phase of mPCa (mHSPC) in 2015 [7]. Several androgen-receptor signaling inhibitors (ARSi)—abiraterone, enzalutamide, and apalutamide—were then approved for the treatment of mHSPC [8].

Although the aforementioned randomized trials showed significant survival improvements in the first- and second-line of mCRPC, the real-world survival benefit in the population of patients outside of clinical trials is largely unexplored. The ideal population of patients enrolled in clinical trials might overestimate the true benefit induced by approved drugs in the general population of patients with newly diagnosed mPCa. For example, not all patients can receive chemotherapy. Although no specific advice is included in the U.S. National Comprehensive Cancer Network guidelines, the European Association of Urology guidelines recommend that docetaxel should be only offered to mHSPC patients who are fit enough for chemotherapy [9]. Of note, the STAMPEDE trial of docetaxel in mHSPC only included patients fit for chemotherapy and without significant cardiovascular history. Many patients with mPCa in the real-world are elderly with many comorbidities, and they cannot receive chemotherapy [10]. In addition, patients with poor general conditions or poor performance status are often not suitable for aggressive anticancer therapies. Moreover, although some retrospective data have been reported [11], no randomized trial has ever assessed the long-term, cumulative benefit on survival that can derive from the temporal sequence of different treatment strategies. Finally, the U.S. insurance policies or limited access to healthcare services could contribute to producing a discrepancy between the expected survival gain and the real-world data [12].

Here, we investigated the survival trends and prognostic variables in patients with de novo mPCa included in the U.S. Surveillance, Epidemiology, and End Results (SEER) database. Given the introduction of chemotherapy in 2004 and of ARSi in 2011, we hypothesized that a significant difference in OS and cancer-specific survival (CSS) was detectable in patients diagnosed in three time periods: 2000–2003 (Cohort A), 2004–2010 (Cohort B), and 2011–2014 (Cohort C). Of note, our study should not be intended to provide data on the efficacy of the newer treatments, but to provide epidemiological results about the survival trends in patients with de novo mPCa diagnosed in the United States in the last two decades.

## 2. Results

### 2.1. Study Cohort

Our selection criteria identified 26,434 patients with de novo mPCa diagnosed between 2000 and 2014. Of these, 6047 were diagnosed between 2000 and 2003 (Cohort A), 11,815 between 2004 and 2010 (Cohort B), and 8572 between 2011 and 2014 (Cohort C). The main characteristics of the study population are summarized in Table 1. Overall, 68.3% of patients were ≥65 years. The percentage of patients younger than 75 years was higher in Cohort C compared to Cohorts B and A (64.8% vs. 61.1% vs. 58.3%, respectively). The majority of patients were white (62.7%), followed by black (19.4%) and Hispanic (11.6%). Metastatic classification (American Joint Committee on Cancer (AJCC), 6th edition) was available for Cohorts B and C. The majority of patients were M1b (72.7%), with a significant difference between Cohorts B (70.1%) and C (76.4%). The full contingency table with the comparison of baseline characteristics among the cohorts is available in Appendix A. The median follow-up was 25, 26, and 29 months in Cohorts A, B, and C, respectively, with a median follow-up of censored patients of 60, 60, and 51 months.

### 2.2. Clinical Outcome and Prognostic Variables

In the 26,434 patients analyzed for OS, the median values for OS in Cohorts A, B, and C were 26 (95% confidence interval (CI) 25.0–27.0), 26 (95% CI 25.3–26.7), and 30 (95% CI 29.1–30.9) months (Figure 2A). In the 26,032 patients analyzed for CSS, the median values of CSS were 31 (95% CI 29.7–32.3), 31 (95% CI 30.1–31.9), and 35 months (95% CI 32.4–33.6) in Cohorts A, B, and C, respectively (Figure 2B). The detailed age-standardized 1- to 5-year OS and CSS are shown in Figure 3.

Age, race, and metastatic stage (the latter was only analyzed in Cohorts B and C) were identified as significant prognostic factors at univariate analysis (data not shown) and were included in the multivariable models.

### 2.3. Multivariable Models

The multivariable models for OS and CSS showed a substantially increased risk of death according to age, with the highest risk in patients ≥85 (Table 2 and Table 3). Black patients showed a slightly higher risk of death compared to white, whereas Asians/Pacific Islanders showed better outcomes compared to white. A 9% decreased risk of death and an 8% decreased risk of cancer-specific death were found in Cohort C compared with Cohort A (hazard ratio (HR): 0.91 (95% CI 0.87–0.95), *p* < 0.001 for OS; HR: 0.92 (95% CI 0.88–0.96), *p* < 0.001 for CSS), whereas no statistically significant differences in OS and CSS were found between Cohorts A and B. Exploratory multivariable models were also performed in Cohorts B and C to include the metastatic stage classification (AJCC, 6th edition), which was found to be associated with distinct OS and CSS outcomes (Appendix A). In these multivariable models, significant OS and CSS advantages were reported in Cohort C compared with Cohort B (HR: 0.94 (95% CI 0.91–0.97), *p* = 0.001 for OS; HR: 0.89 (95% CI 0.85–0.92), *p* < 0.001 for CSS). In the exploratory subgroup analysis comparing the OS and CSS of Cohort C with Cohort B, a significant interaction was found among the subgroups of the AJCC metastatic classification. More pronounced OS and CSS advantages in Cohort C were shown in M1c patients compared with patients with metastases that were limited to nodes or bone (M1c HR: 0.87 (95% CI 0.81–0.94), interaction *p* = 0.014 for OS; M1c HR: 0.81 (0.75–0.88), interaction *p* = 0.015 for CSS) (Table 4).

## 3. Discussion

Several randomized trials demonstrated that both chemotherapy and ARSi provided a significant survival benefit in mPCa [1,2,3,4,5,6,7,8]. However, the real-world survival outcomes of patients with de novo mPCa remain largely unexplored.

A recent analysis compared 590 patients with mCRPC, who were diagnosed and treated in two treatment eras (2004–2007 vs. 2010–2013) at the Dana–Farber Cancer Institute [11]. The authors demonstrated a 41% decreased risk of death in the newer treatment era, with a median OS gain of 6 months. In addition, the cumulative benefit from the newer therapies was more pronounced in longer-term survivors and de novo patients. Although this study provided useful information, all patients had castration-resistant disease, only 216 had de novo mPCa, and they were all managed in a top-level institution.

In another study, Helgstrand and colleagues analyzed the incidence and mortality data of patients with de novo mPCa included in the SEER database and in the Danish Prostate Cancer Registry [13]. In patients diagnosed between 2000 and 2009, the median OS was 22 months in SEER and 30 months in the Danish Registry. The five-year overall mortality was 80.0% in both registries in the period of 2000–2004, remained stable (80.5%) according to SEER in 2005–2008, and decreased to 73.2% according to the Danish Registry in 2005–2009.

Although the monocentric experience of the Dana–Farber Cancer Institute and the Danish data confirmed the potential survival gain offered by newer treatments, the SEER analysis by Helgstrand and colleagues did not show substantial survival changes after 2004.

In the present SEER-based analysis, we investigated whether the introduction of both chemotherapy and ARSi in mCRPC had substantially changed the real-world OS and CSS in the population of patients with de novo mPCa diagnosed in the United States of America in three different time periods (2000–2003—Cohort A, 2004–2010—Cohort B, 2011–2014—Cohort C). Although the patients were allocated to these cohorts regardless of having received a specific treatment, we highlight that docetaxel was approved by the FDA for the treatment of mCRPC in 2004, whereas ARSi was approved from 2011 onwards (Figure 1).

More than 26,000 patients diagnosed between 2000 and 2014 were included in our analysis; of these, 6047 were allocated to Cohort A, 11,815 to Cohort B, and 8572 to Cohort C (Table 1). We found that age had a significant impact on patients’ OS and CSS (Table 2 and Table 3). In the multivariable model, patients older than 85 showed a double risk of dying compared with patients between 15 and 54 years old, and the hazard ratio for death was also significantly unfavorable in patients aged 75–84. Although this figure might be at least in part attributable to the reduced expected survival, older patients may also be less likely to receive the same treatments as their younger counterparts, especially chemotherapy.

We did not find a significant difference in the OS and CSS between Cohort A and Cohort B (Figure 2). Conversely, we observed a statistically significant improvement in the OS and CSS of patients included in Cohort C, who showed a decreased risk of death of 9%, a decreased risk of cancer-specific death of 8%, and a median OS gain of 4 months compared with Cohort A. The comparison of Cohort C with Cohort B, adjusted for the metastatic stage, also demonstrated an OS improvement of 6% and a CSS improvement of 11%. When compared with the other metastatic stages, we found that patients with M1c disease showed the worst survival, but had a more pronounced OS and CSS improvement in the newer ARSi era compared with M1a or M1b patients (Table 4). Although the reason for this observation remains unknown, the presence of visceral metastases might lead to more aggressive pharmaceutical approaches and more adherence to treatment that could result in increased benefit compared with the other stages.

The median OS gain of chemotherapy and ARSi in randomized trials for mCRPC was 2–4 months in first-line [1,5,6] and 4–5 months in second-line [3,4]. Although our study was not designed to demonstrate the potential benefit of chemotherapy or ARSi, a more robust OS and CSS improvement would have been expected in patients diagnosed in 2011–2014, after the introduction of several agents in clinical practice (Figure 1). A median OS improvement of 4 months in Cohort C compared with Cohort A appears to be quite discouraging. Regardless of cohort analysis, the probability of survival after 3 years from diagnosis was 40.0% in 2000 and 46.8% in 2014 (Figure 3). Similarly, the five-year probability of survival was 24.0% in 2000 and 28.2% in 2012. Several reasons might explain these disappointing results.

First, the degree of benefit seen in clinical trials does not necessarily translate into the real-world setting. Screen failure rates on trials are relatively high and can easily affect the ultimate generalizability of trial results to the real-world population.

Second, our study was based on patients diagnosed with de novo mHSPC who were supposed to receive androgen-deprivation therapy (ADT) as a first-line treatment for metastatic disease, and subsequently docetaxel or ARSi as a first-line treatment for mCRPC. The number of patients who died without receiving a first-line treatment for mCRPC or refused therapies for mCRPC was unknown. The information on the number of lines of treatment, type of treatment, disease burden, number and site of metastases, body mass index, performance status, and comorbidities was not available in the SEER database, and these potential confounders were not included in our analysis. In addition, we acknowledge that some patients could have received chemotherapy or ARSi outside of the defined cohort allocation in the context of clinical trials or some years after mPCa diagnosis.

Third, the medical costs and the health insurance policies might have significantly reduced the extensive use of ARSi and chemotherapy in the general population of patients with de novo mPCa diagnosed and treated in the United States, affecting their survival outcomes. Ramsey and colleagues reported that the cumulative incidence of bankruptcy in the first 5 years after prostate cancer diagnosis is 38% (nearly 50% in metastatic stage), and the risk of mortality is almost twice as high among patients with prostate cancer who file for bankruptcy compared with those who do not [12]. Further studies should investigate whether insurance policies or limited access to healthcare services could contribute to such disappointing survival gains observed in the SEER registry after the introduction of chemotherapy and ARSi.

Fourth, patients with de novo mPCa showed worse time to castration and survival compared with those who relapsed after local therapy, irrespective of treatment received [14,15]. Therefore, the intrinsic aggressiveness of de novo mPCa could have also led to decreased survival gains in this patient population. Although discouraged by international guidelines in recent years, possible premature discontinuation of ARSi and chemotherapy based on PSA progression without clinical or radiographic progression could have also affected the outcome data of patients diagnosed between 2004 and 2014 [16].

Finally, we acknowledge that our study excludes the possible benefit induced by docetaxel or ARSi in mHSPC, given their approval for this setting in the latest years (Figure 1). The earlier use of these agents provided OS gains that exceeded 12 months in randomized trials for mHSPC [8]. Future analyses could also detect additional survival benefits that might be provided by an increased knowledge in the sequencing of agents for mCRPC and by the biomarker-driven selection of patients suitable for specific drugs (i.e., poly (ADP-ribose) polymerase (PARP) inhibitors) [17,18,19].

## 4. Patients and Methods

The SEER*Stat software was used to select all patients with de novo mPCa from the SEER Research Data 2000–2017 [20]. Patients were assigned to three cohorts based on the year of diagnosis (2000–2003: Cohort A; 2004–2010: Cohort B; 2011–2014: Cohort C). Patients with prostate cancer were identified using the codes for malignant adenocarcinoma (8140/3) and prostate gland (C61.9). Only patients with a single tumor in medical history were selected. Metastatic patients were identified using a combination of the American Joint Committee on Cancer (AJCC) classification from the 3rd and 6th editions. According to the November 2019 submission of SEER data, the study cut-off for survival data was 31 December 2017. In order to minimize potential bias related to different follow-up among the cohorts, the maximum follow-up was fixed to 5 years, and patients diagnosed from 2015 onwards were excluded. OS was defined as the time from mPCa diagnosis to death from any cause. CSS was defined as the time from mPCa diagnosis to death from prostate cancer. Patient age (SEER standard for survival in prostate cancer: 15–54, 55–64, 65–74, 75–84, 85+), race, year of mPCa diagnosis, metastatic stage, and outcome data were included in the case listing session of SEER*Stat. The variables described were analyzed in univariate analysis using Kaplan–Meier curves and a log-rank test. A *p*-value ≤ 0.05 was considered statistically significant. Cox proportional hazards models were used to test the effects of covariates on OS and CSS. Only patients who had known values for the variables of interest were included. The chi-square statistic was applied to compare groups. The IBM software Statistical Package for Social Sciences (SPSS) Version 23 and RStudio Version 1.2.5001 were used for data analysis.

## 5. Conclusions

Our large-scale, retrospective study suggested that the real-world OS and CSS have not drastically changed during the last two decades in patients with de novo mPCa diagnosed in the United States. The median OS of these patients remained poor and did not exceed 2.5 years. Although we acknowledge that several potential confounding factors have not been adjusted in our analysis, our study highlighted that a significant discrepancy might exist between the benefit observed in randomized trials and the real-world data. Several reasons might explain this discrepancy, such as a lack of access to cancer cares, patients’ ineligibility or refusal of treatments, insurance issues, or intrinsic aggressiveness of de novo disease. However, given that patients were not allocated according to the receipt of specific treatments, our results should not be used to draw conclusions about the potential efficacy of systemic therapies.

## Figures and Tables

**Figure 1 cancers-12-02855-f001:**
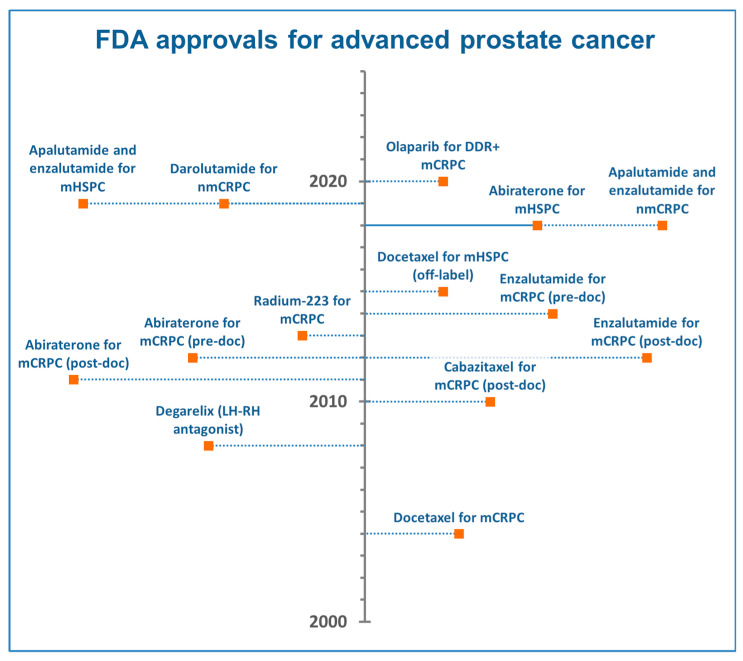
Regulatory timeline of approvals in advanced prostate cancer therapies. DDR+: DNA damage response genes mutated; mHSPC: metastatic hormone-sensitive prostate cancer; mCRPC: metastatic castration-resistant prostate cancer; nmCRPC: nonmetastatic castration-resistant prostate cancer; post-doc: post-docetaxel; pre-doc: pre-docetaxel.

**Figure 2 cancers-12-02855-f002:**
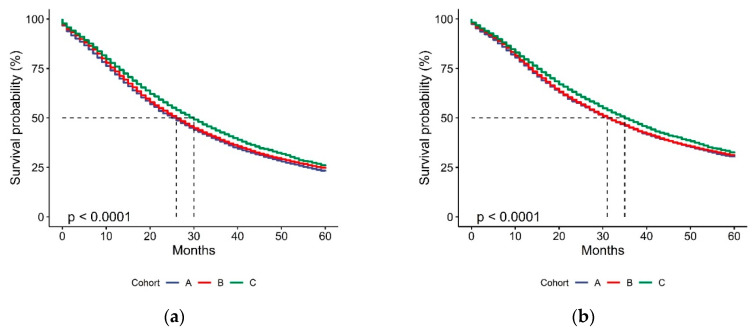
Kaplan–Meier estimations of overall survival (OS) (**a**) and cancer-specific survival (CSS) (**b**) according to cohort allocation. *p*-value from log-rank test.

**Figure 3 cancers-12-02855-f003:**
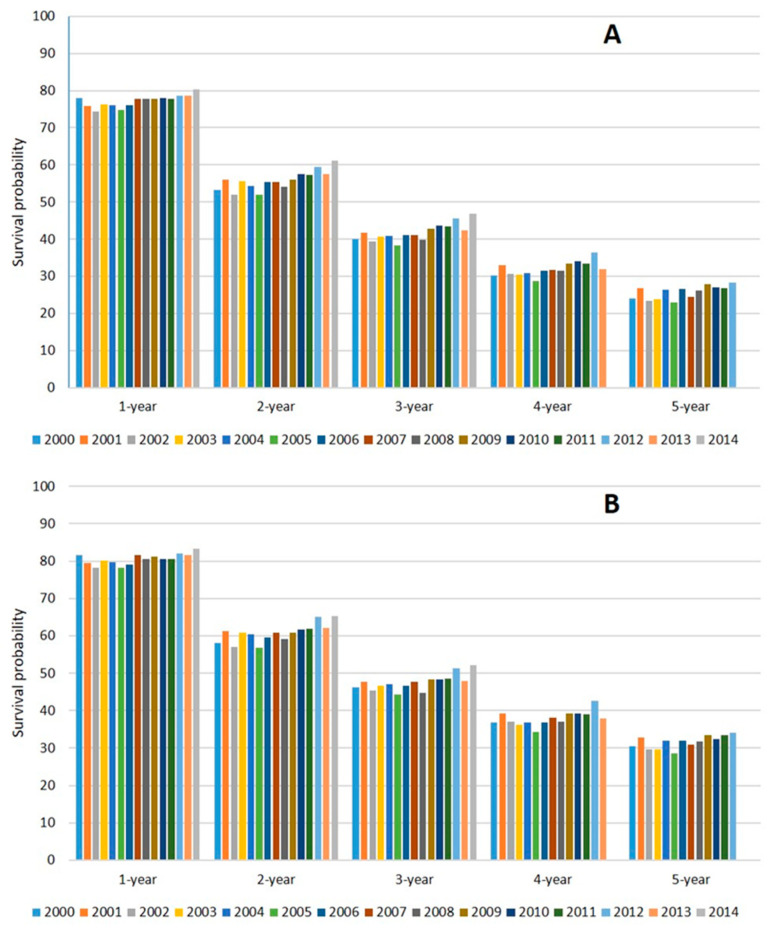
Age-standardized 1- to 5-year OS (**A**) and CSS (**B**) of patients according to year of diagnosis.

**Table 1 cancers-12-02855-t001:** Basal characteristics of patients.

Variables		Number of Patients (%)			
		Total	2000–2003	2004–2010	2011–2014
**Age (years)**	15–54	2087 (7.9)	474 (7.8)	970 (8.2)	643 (7.5)
	55–64	6323 (23.9)	1250 (20.7)	2857 (24.2)	2216 (25.9)
	65–74	7892 (29.9)	1804 (29.8)	3391 (28.7)	2697 (31.5)
	75–84	7099 (26.9)	1862 (30.8)	3268 (27.7)	1969 (23.0)
	≥85	3033 (11.5)	657 (10.9)	1329 (11.2)	1047 (12.2)
	Total	26,434 (100)	6047 (100)	11,815 (100)	8572 (100)
**Race**	White	16,513 (62.7)	3830 (63.5)	7361 (62.5)	5322 (62.3)
	Black	5111 (19.4)	1227 (20.3)	2279 (19.3)	1605 (18.8)
	Am. Indian/Alaska Native	170 (0.6)	31 (0.5)	76 (0.6)	63 (0.7)
	Asian or Pacific Islander	1484 (5.6)	329 (5.4)	680 (5.8)	475 (5.6)
	Hispanic	3066 (11.6)	614 (10.2)	1377 (11.7)	1075 (12.6)
	Total	26,344 (100)	6031 (100)	11,773 (100)	8540 (100)
**Metastatic stage**	M1a	1097 (5.6)	-	610 (5.3)	487 (5.9)
	M1b	14,301 (72.7)	-	8011 (70.1)	6290 (76.4)
	M1c	4265 (21.7)	-	2811 (24.6)	1454 (17.7)
	Total	19,663 (100)	-	11,432 (100)	8231 (100)

**Table 2 cancers-12-02855-t002:** Multivariable analysis for OS.

Variables		Number of Patients	HR	95% CI		*p*
				Lower	Upper	
**Age (years)**	15–54	2081				<0.001
	55–64	6300	0.98	0.92	1.04	0.515
	65–74	7857	1.03	0.97	1.09	0.286
	75–84	7078	1.42	1.34	1.50	<0.001
	≥85	3028	2.18	2.04	2.32	<0.001
**Race**	White	16,513				<0.001
	Black	5111	1.10	1.06	1.14	<0.001
	Am. Indian/Alaska Native	170	1.08	0.91	1.28	0.393
	Asian or Pacific Islander	1484	0.74	0.69	0.79	<0.001
	Hispanic	3066	0.94	0.90	0.98	0.010
**Year of diagnosis**	2000–2003 (Cohort A)	6031				<0.001
	2004–2010 (Cohort B)	11,773	0.97	0.94	1.01	0.145
	2011–2014 (Cohort C)	8540	0.91	0.87	0.95	<0.001

**Table 3 cancers-12-02855-t003:** Multivariable analysis for CSS.

Variables		Number of Patients	HR	95% CI		*p*
				Lower	Upper	
**Age (years)**	15–54	2049				<0.001
	55–64	6216	0.94	0.88	0.99	0.048
	65–74	7720	0.93	0.88	0.99	0.033
	75–84	6979	1.20	1.12	1.27	<0.001
	≥85	2987	1.74	1.62	1.87	<0.001
**Race**	White	16,376				<0.001
	Black	5053	1.09	1.04	1.13	<0.001
	Am. Indian/Alaska Native	167	1.01	0.83	1.23	0.922
	Asian or Pacific Islander	1423	0.73	0.67	0.78	<0.001
	Hispanic	2932	0.95	0.91	1.00	0.076
**Year of diagnosis**	2000–2003 (Cohort A)	5928				<0.001
	2004–2010 (Cohort B)	11,599	0.99	0.95	1.03	0.596
	2011–2014 (Cohort C)	8424	0.92	0.88	0.96	<0.001

**Table 4 cancers-12-02855-t004:** Subgroup analysis of OS and CSS between Cohorts C and B.

2011–2014 (Cohort C) vs. 2004–2010 (Cohort B)		Number of Patients	HR	95% CI		*p*
				Lower	Upper	
OS Metastatic Stage	M1a	1088	1.09	0.93	1.28	0.014 *
	M1b	14,250	0.96	0.92	0.99	
	M1c	4254	0.87	0.81	0.94	
All ^1^		19,592	0.94	0.91	0.97	0.001
CSS Metastatic Stage	M1a	1069	1.01	0.85	1.20	0.015 *
	M1b	14,050	0.91	0.87	0.95	
	M1c	4189	0.81	0.75	0.88	
All ^1^		19,308	0.89	0.85	0.92	<0.001

Multivariable models including age and race were used to compute the hazard ratios (HR) and their 95% confidence intervals (CI) for OS and CSS in the metastatic subgroups of patients diagnosed in 2011–2014 vs. 2004–2010. * *p*-value for interaction; ^1^ Multivariable model including age, race and metastatic stage for OS and CSS (Cohort C vs. Cohort B).

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
