# Peer review of "Epidemiological Characteristics and Survival in Patients with De Novo Metastatic Prostate Cancer"

_cancers, 2020, doi:10.3390/cancers12102855_

Round 1

Reviewer 1 Report

Thanks for the comments. I would like to make a remark based on your comments to reviewer 2 and myself. You stated that "Of note, our study is not supposed to provide information about the efficacy of systemic treatments, but to report the real-life survival times over time in the overall population of patients with newly diagnosed metastatic prostate cancer".

It that is the aim of the manuscript, please make it more clear in the introduction and methods of the paper. I understood that you aimed to compare survival based on treatment allocation given the classification used to select the cohorts in the paper. I will advise to make less emphasis in the treatment pathways and more in the time frames to make it more clear and accurate. 

I would also extend the paragraph about limitations including all the potential confounders not available in SEER but potentially related with the two endpoints studied in the paper.

Author Response

Thanks for the comments. I would like to make a remark based on your comments to reviewer 2 and myself. You stated that "Of note, our study is not supposed to provide information about the efficacy of systemic treatments, but to report the real-life survival times over time in the overall population of patients with newly diagnosed metastatic prostate cancer".

It that is the aim of the manuscript, please make it more clear in the introduction and methods of the paper. I understood that you aimed to compare survival based on treatment allocation given the classification used to select the cohorts in the paper. I will advise to make less emphasis in the treatment pathways and more in the time frames to make it more clear and accurate.

I would also extend the paragraph about limitations including all the potential confounders not available in SEER but potentially related with the two endpoints studied in the paper.

RE: As suggested, we added a statement at the end of introduction that clearly explains the aim of our study. We also completely revised the discussion to make it clearer and we extended the paragraph on potential confounders not included in SEER database.

Reviewer 2 Report

This study addresses an important question of what happens in real life for patients with de novo metastatic prostate cancer and has the advantage of covering a very large number of American patients.

Overall, the article suffers from an overly technical description which prevents the reader from clearly understanding the message of this analysis. By explaining the results more simply as in the responses to the reviewers, the objective and the conclusions of this study are more appealing.

In the abstract, the term M1c should be explained and the message might be more explicit, such as in the response to reviewer 2 “(in the real-life), a median patient diagnosed in 2014 shows a survival gain that does not exceed 4 months compared to one patient diagnosed in 2000. » instead of the last sentence.

To be clearer and to avoid confusion, the goal of the study should be mentioned at the end of the introduction, as in the response to reviewer 1: “Our study should not be intended to provide data on the efficacy of the newer treatments, but to provide epidemiological results about the survival trends in patients with de novo metastatic prostate cancer in the last two decades. »

In the result section, the statistical difference between the 3 groups should be mentioned and showed in supplementary data, thus justifying the choice of the multivariable models used.

Because the lack of information on the type of treatment received by patients is whatever frustrating, the authors should stress that OS is little different between groups, regardless of treatment, in the same way they have responded to reviewer 3 for figure 3.

Author Response

This study addresses an important question of what happens in real life for patients with de novo metastatic prostate cancer and has the advantage of covering a very large number of American patients.

Overall, the article suffers from an overly technical description which prevents the reader from clearly understanding the message of this analysis. By explaining the results more simply as in the responses to the reviewers, the objective and the conclusions of this study are more appealing.

RE: As suggested, we significantly modified the discussion and conclusions in order to make it more readable, linear and understandable. The simple abstract could help to this purpose. We also added a more explicit analysis that directly compares the 2-year survival probability in 2000 and in 2014 and the 5-year survival probability in 2000 and 2012.  

In the abstract, the term M1c should be explained and the message might be more explicit, such as in the response to reviewer 2 “(in the real-life), a median patient diagnosed in 2014 shows a survival gain that does not exceed 4 months compared to one patient diagnosed in 2000. » instead of the last sentence.

RE: The abstract has been modified according to this comment.

To be clearer and to avoid confusion, the goal of the study should be mentioned at the end of the introduction, as in the response to reviewer 1: “Our study should not be intended to provide data on the efficacy of the newer treatments, but to provide epidemiological results about the survival trends in patients with de novo metastatic prostate cancer in the last two decades. »

RE: As requested, we specified the goal at the end of introduction.

In the result section, the statistical difference between the 3 groups should be mentioned and showed in supplementary data, thus justifying the choice of the multivariable models used.

RE: We added the S1 table which compares the differences among the 3 groups with appropriate statistical analysis.

Because the lack of information on the type of treatment received by patients is whatever frustrating, the authors should stress that OS is little different between groups, regardless of treatment, in the same way they have responded to reviewer 3 for figure 3.

RE: This point is now discussed in the paragraph that we added to the discussion session. The direct comparison of the 2-year survival probability in 2000 and 2014 and of the 5-year survival probability in 2000 and 2012 have been implemented.

This manuscript is a resubmission of an earlier submission. The following is a list of the peer review reports and author responses from that submission.

Round 1

Reviewer 1 Report

The present retrospective analysis aims to evaluate the real-world survival benefit obtained with the introduction of novel therapeutic strategies for newly diagnosed metastatic prostate cancer (mPCa). The authors hypothesized that a the approval of docetaxel, first, and ARSi, then, could have significally modified the outcomes of patients diagnosed in 3 treatment eras: pre-docetaxel, docetaxel and ARSi.

Since the real-world survival outcomes of patients with mPCa remain largely unexplored, the present analysis raises interest for the clinicians involved in PCa care. Unfortunately, the use of the SEER database (that provides epidemiological data), did not conset to provide deep conclusive considerations about the results of the analysis due to the long series of data missing (type, number, sequence of treatments that patients receveid, for example). This is an important limit that the Authors detailed in the discussion section.

The paper is clear and well written. The results are clearly presented and supported by methods applied and background provided.

No revisions are needed.

Reviewer 2 Report

This retrospective analysis seeks to evaluate the overall survival and cancer-specific survival of over 26 thousands patients diagnosed with de-novo metastatic castration-sensitive prostate cancer. The authors should be commended for the effort of conducting this study on such a large sample.

However, I raise the following issues:

  • This report lacks of originality as other reports in the recent past investigated the OS of patients with de-novo mCSPC (i.e. Francini et al. The Prostate 2018).
  • The report tends to be poorly written all along.
  • Introduction is poor. In particular: a) the reason for choosing to explore OS and CSS of patients with de-novo disease was not described. b) I take issue with this sentence "the potential cumulative benefit on survival that can derive from the temporal sequence of different treatment strategies is currently unknown". As authors later reported in the discussion, this was explored by Francini et al. in a recent retrospective analysis. c) Reason for picking CSS as primary endpoint should have been described.
  • Design is biased. In particular: the categorization into 3 cohorts lacks of logic when analyzing the time frames. Because docetaxel was approved for mCRPC in 2004 and the five newer therapies for mCRPC received FDA approval between 2010 and 2014, a portion of patients in cohort A (2000-2003) probably had docetaxel as those in cohort B (2004-2010) and some in cohort B likely had some of the newer therapies as those in cohort C (2011-2014), during the course of their disease. That introduces relevant clinical biases. A more efficient classification could have been 2000-2007 and 2010-2014. 
  • Methods: a) Neither OS nor CSS were defined. b) Age is typically reported as a continuous variable. No explanation about the numerical categorization or the chosen cut-off points was given. c) MVA lacks inclusion of relevant clinical factors known to be associated with OS (i.e. ECOG PS, baseline pain, baseline PSA, etc.).
  • Results are not well exposed i.e. "In the 26032 patients analyzed for CSS, the median values of CSS were 31 (95% CI, 29.7-32.3), 31 (95% CI, 30.1- 72 31.9) and 35 months (95% CI, 32.4-33.6)" which cohort has what CSS? 
  • Discussion: a) "Therefore, our data suggest the significant efforts should be spent to specifically improve the outcomes of the older and frail population of patients with mPCa." I see no correlation between this general statement and the previous observation that mPCa patients with age 15-54 have longer survival than patients aged 75-84, and it should be deleted b) A limit which was not mentioned is that patients in cohort A are generally older than those in B and C which may have biased results.

Reviewer 3 Report

I really like the research idea presented in the manuscript, the impact of the different drug regimens implemented in the last decade for the novo metastatic prostate patient in survival outcomes (OS and CSS) in a real world setting from the SEER database.

I have few comments on the manuscript itself:

Abstract – please state more clearly the aim as it is not clear.

Introduction – clear maybe more information regarding the guidelines around the suitability of the treatments based on the specific patient profile will be interesting. Together with some more patient profiling information.

A figure with a timeline plotting the approval dates of the different drug regimes and the selected cohorts could improve the understanding of the aim.

Results

Study cohort – what is the selection criteria?

Table 1 – could be interesting to assess if the groups are statistically significantly different for the 3 variables reported.

Clinical outcome and prognostic variables -  Age, race and metastatic stage (this latter was only analysed in cohort B and C) were identified 75 as significant prognostic factors at univariate analysis – which univariate analysis?

Multivariate analysis - The results on age and race seem consistent with previous literature. The protective effect seen mainly in cohort C in respect to the older cohort A is interesting, but by itself cannot respond to the question whether the metPCa drugs have a true benefit for patient in terms of survival in a real world setting  given the limitation on data available for patients such as drug regimes, comorbidities, etc... Moreover, the protective effect on the M1c group in the sensitivity analysis is not clearly explain.

Limitations

A more complete phenotyping exercise of the cohorts will have provided a better insight on the potential benefit of the different drugs available for de novo metPCa patients in a real-world setting. The selection of the cohorts based solely on year of diagnosis seems a bit arbitrary to me given that the drug regime they followed is unknown. Moreover, Age, race and staging are not holding enough information to assess the potential difference between the cohorts responsible for the impact on the patient’s OS and CSS. A number of other factors are potentially associated with the OS and CSS of the patients, as for example the specific treatment regimes, BMI, lifestyle, comorbidities, etc are potential confounders that should have been included in the study.

Overall, I think the aim of the paper is very relevant and there is a gap on the field to assess the true benefit on the metPCa drug regimes. However, I think the study design failed to include many other variables that are key to assess the potential benefit of the drugs such a more detailed phenotype of the patients and moreover the drug regime those patients followed. The manuscript guides us towards the right direction to answer the research question, but I will recommend to include the above mention variables to be able to understand the true impact of the drug regimes on those patients.